# Determination of the Simplified Daylight Glare Probability (DGPs) Criteria for Daylit Office Spaces in Thailand

**Kittiwoot Chaloeytoy [1],*** , **Masayuki Ichinose [1]** and **Szu-Cheng Chien [2]**

[1]  Department of Architecture and Building Engineering, Graduate School of Urban Environmental Sciences, Tokyo Metropolitan University, 1-1, Minami Osawa, Hachioji-shi, Tokyo 192-0397, Japan; ichinose@tmu.ac.jp

[2]  Engineering Cluster, Singapore Institute of Technology, 10 Dover Drive, Singapore 138683, Singapore; SzuCheng.Chien@singaporetech.edu.sg

*   Correspondence: kittiwoot-chaloeytoy@ed.tmu.ac.jp; Tel.: +81-50-6864-5216

**Abstract:** The increasing trend of employing glazed façades to utilize daylight in the buildings has made it necessary to develop measures to avoid excessive sunlight penetration in such daylit spaces. In Thailand, only a few studies have focused on daylight glare, and therefore, applicable criteria are required to fulfill the local preference. This study aimed to determine daylight glare thresholds on the basis of the occupants' responses. A post-occupancy evaluation with a simplified daylight glare probability (DGPs) model was performed in eight open-plan office spaces located in Bangkok, Thailand. The occupants participated in a survey including a subjective questionnaire; the results showed that the DGPs model performed effectively for glare prediction, with a preference for a lower level than that found in the current references. Statistical analysis helped mark the threshold values for each glare sensation level: imperceptible–perceptible = 0.22; perceptible–disturbing = 0.24; and disturbing–intolerable = 0.26. The findings of this study can be considered as initial evidence for improving the understanding of local occupants' perspectives and illumination standards, which currently encourage daylight utilization without any specific glare control strategies.

**Keywords:** discomfort glare; daylight glare; vertical illuminance; simplified daylight glare probability; office space; post-occupancy evaluation; tropical regions

## 1. Introduction

As the trend of glazed façades continues to increase [1], the façades themselves remain the main source that can cause negative effects on buildings. Being one of the most vulnerable elements, glazed façades have usually represented a critical aspect in building design because, if not properly conceived, they can have an impact on energy demands. Moreover, their adoption can translate into occupants' comfort [2,3]. Over the last decades, strategies to improve human health and facilitate well-being in buildings have been at the forefront of building practices [4].

Office buildings are gaining popularity globally, and particularly in Southeast Asia. In Thailand, growth in new office spaces is predicted, owing to the high regional economic growth [5]. These newly constructed offices are expected to contribute positively to their inhabitants by assessing the occupants' comfort [6]. Green building certification, which ascertains the well-being standard, has evolved with an indoor environmental concept alongside its profound effects on natural environments and building occupants. Concerning the glazed façade effect, the transmitted solar radiation results in a large amount of solar heat gain and daylight entering simultaneously, which has been described in the sensory terms of thermal comfort and visual comfort. The achievement of both are important and dominant features in any working situation [7].

To deal with heat load, thermal comfort evaluation corresponding to the transmitted solar radiation is specified as a predicted mean vote (PMV) index with its product as predicted percentage of dissatisfied (PPD) [8,9]; these are reference methods for obtaining several thermal comfort criteria in building rating systems, including local standards [10]. Thermal sensation can be predicted from the experience of an individual with physiological factors and thermal variables. Visual comfort is specified in terms of the quality of interior lighting—artificial light with daylight—as important building features [7]. Daylight is introduced to reinforce circadian rhythms and reduce electrical lighting load [4,9]. As highly glazed buildings allow a large amount of daylight into interior spaces, overestimated brightness can cause visual fatigue due to discomfort glare issues [11]. Thus, visual comfort in the interior spaces is commonly restricted by evaluation of the discomfort glare index [12]. However, in terms of Thai standards, its effects are opaquely mentioned without consideration of occupants' comfort [13], although various studies have been conducted to predict visual comfort in such daylit environments [14]. Without an appropriate strategy, the lack of applicable criteria may control discomfort glare purposelessly. Therefore, it is necessary to recognize evaluation of discomfort glare as a key step in achieving visual comfort and to adopt steps for improving occupants' comfort. These following backgrounds are mentioned to overview holistic concept of discomfort glare, concerning its review under the contextual study, with acronym following Table 1.

**Table 1.** Acronym list.

| | |
|---|---|
| $\omega_{s,i}$ | Solid angle |
| $L_{s,i}$ | Luminance (cd/m$^2$) |
| $P_i$ | Guth position index for the field of view |
| DGI | Daylight glare index |
| DGP | Daylight glare probability |
| DGPs | Simplified daylight glare probability |
| Ev | Vertical illuminance (lux) |
| HDR | High dynamic range |
| IGU | Insulated glass unit |
| PMV | Predicted mean vote |
| PPD | Predicted percentage of dissatisfied (or discomfort) |
| SHGC | Solar heat gain coefficient |
| UGR | Unified glare rating |
| VLT | Visible light transmittance |
| WWR | Window-to-wall ratio |

### 1.1. Application of the Simplified Daylight Glare Probability (DGPs)

The use of daylight has been suggested as a building design strategy and an operational method to improve work performance [7]. Thus, it has become crucial to provide appropriate conditions and avoid excessive sunlight penetration, which can cause visual discomfort to the occupants. Daylight glare index (DGI) [15] and daylight glare probability (DGP) [16] are well-known glare indices that are used to assess daylit spaces for large-area sources of daylight glare such as glazed façades. However, DGP performs better than DGI, especially in bright scenes with daylight. Wymelenberg et al. [17] evaluated visual comfort in a daylit office laboratory environment in the USA and found that DGP has a stronger correlation with subjective glare sensation than DGI. Jakubiec et al. [18] found that DGP yields the most plausible results compared to other visual comfort indices. Suk et al. [19] showed that

DGP outperformed other discomfort glare indices with reference to responses of occupants in office spaces. The DGP equation (Equation (1)) can be calculated as

$$DGP = 5.87 \times 10^{-5} \, Ev + 9.8 \times 10^{-2} \log\left(1 + \sum_i \frac{L_{s,i}^2 \, \omega_{s,i}}{Ev^{1.87} \, P_i^2}\right) + 0.16 \tag{1}$$

where $L_{s,i}$ is the source luminance (cd/m$^2$), $\omega_{s,i}$ is the solid angle, Ev is the vertical illuminance (lux), and $P_i$ is the Guth position index for the field of view. The DGP developed by Wienold and Christoffersen [16] is a practical index used to evaluate daylight glare in daylit spaces; it was studied on the basis of experimental data obtained from a test room of a private office space involving human test subjects. The results indicated statistically strong correlations between the subjective votes and the metric on the basis of the distribution of luminance and vertical illuminance, considering the effect of both on the eye of the observer. However, Wienold proposed the concept of simplified daylight glare probability or DGPs [20], which considered only vertical illuminance to predict the metric since it had a strong correlation to the subjective votes; it should be noted that the DGPs model is an accurate metric when there is no direct sunlight. The equation (Equation (2)) is expressed as

$$DGPs = 6.22 \times 10^{-5} Ev + 0.184 \tag{2}$$

In general, illuminance-based data was commonly addressed in discomfort glare evaluation or interior lighting standards [21]. Lighting design metrics could be simply provided by horizontal illuminance on the desktop level due to its ease of use and prevalence in practice [17,22]. It is generally mentioned in terms of useful daylight illuminance (UDI), which determines when daylight illuminance is useful for the occupants [23]. Mardaljevic [24] recommended the division of achieved UDI where supplementary electrical lighting may be used for the daylight illuminance levels from 100–500 lux, while daylight alone is sufficient for the illuminance levels from 500–2000 lux. Reinhart et al. [25] and Nabil et al. [26] found that the work plane illuminance was suggested below 2000 lux for achieving visual comfort. However, the light sources were located above the line of sight. The quantity of light registered at the eye level should be highlighted to represent discomfort glare perceived by occupants [17,20,22]. Bellia et al. [27] studied daylit offices and noted that the vertical illuminance at the eye level was different from that measured at the work plane level owing to the amount daylight entering with the buildings' characteristics, showing that the definition of visual discomfort could be mentioned on the basis of the vertical illuminance. Annual visual discomfort frequency [28] suggested that the vertical illuminance should be lower than 2670 lux. A filed study by Suk [29] on discomfort glare inside daylit offices showed that vertical illuminance was considered an acceptable proxy of human sensation. Thresholds were defined to achieve visual comfort as low as 875–2000 lux for the seated position in the parallel-to-window direction, and below 1479 lux in the facing-to-window direction. Velds [30] showed that the vertical illuminance measurement near the glazed façade was acceptable to monitor visual comfort. Bian et al. [14] and Wymelenberg et al. [17] found that vertical illuminance outperformed the complex metrics to predict glare perception.

As the DGPs was a product of vertical illuminance, Kleindienst and Andersen [31] tested the usability of DGPs and found that it could be considered as a predictor of human sensation when discomfort glare was mostly caused by the quantity of light hitting the eye. Karlsen et al. [32] studied the verification of DGPs alongside illuminance-based data with human sensation in the test room, finding that they were highly responsive for glare sensation of the test subjects. For the definition of comfort classes, the DGPs model was used in the simulation study of Wienold [20] to determine the threshold values of each. In his study, the results were analyzed with users' assessment from test room study [16]. The suggestions were marked as imperceptible glare for the range below 0.35, perceptible glare for the range from 0.35 to 0.40, disturbing glare for the range from 0.40 to 0.45, and as intolerable glare for the range greater than 0.45.

Since the vertical illuminance dominated subjective responses with its contributions to DGPs, the illuminance-based approach was generally applied to evaluate daylight glare concerning the DGPs model. The simulation study of Giovannini et al. [33] found that this simplified approach to calculate the daylight glare comfort classes using vertical illuminance had a good correlation with DGP evaluation, and it was less time consuming in practical terms. Thus, it is worthwhile to adopt this model for daylight glare evaluation in post-occupancy survey of buildings, and its usability needs to be explored under the contextual study.

*1.2. Daylight Glare Study in Thailand*

In Thailand, daylight access is encouraged along with artificial light to achieve energy saving and provide a sense of naturally positive effects [10,34,35]. For interior lightings, discomfort glare is a referred criterion of the unified glare rating (UGR) index [34]. However, UGR only performs effectively to assess glare from small glare sources such as interior lamps, rather than from large-area sources such as glazed façades [36]. Thus, UGR is unsuitable for daylit spaces, which are commonly found in the urban context of Thailand.

From the above-mentioned background, the DGPs model can be recognized as a notable index due to its strong correlation with subjective responses and concerns for a large-area glare source. However, most of the studies on DGPs have been conducted on Western subjects. Visual sensation in tropical regions is different from that in the West, owing to differences in climate, demography, and culture [37]. Ijspeert et al. [38] found greater intraocular light scattering in blue-eyed people, which is representative of Westerners, compared to that in brown-eyed people such as Southeast Asians. Thus, to optimize the evaluation method, there is a need to develop applicable criteria on the basis of the local occupants' requirements. Ramasoot et al. [13] reviewed illumination guidelines in Thailand and pointed out that the recommended illumination levels found in local standards were unclear. An acceptability-based approach was required to determine the amount of interior lighting considered "acceptable" or "preferable" by occupants.

Studies that focus on DGPs in Thailand are lacking. Meanwhile, those in terms of DGP are rarely mentioned since they originated from the same concept and background. The simulation study of Samithsudanon [39] recommended the installation of shading devices for office buildings in Thailand to achieve visual comfort by consideration of DGP below 0.35. In terms of tropical region studies, field surveys were conducted by Hirning et al. in Malaysia [40] and Mangkuto et al. in Indonesia [37] to determine DGP threshold values. These studies demonstrated a lower threshold value of glare sensation with a preference of DGP below 0.35, which was perceived as comfortable (imperceptible glare) on the basis of current recommendations provided by Wienold [20]. Both collected daylighting variables in occupied spaces using high dynamic range (HDR) images; they then employed software post-processing to generate the DGP. Meanwhile, the glare sensation vote from the occupants was expressed in the questionnaire survey. Physical and subjective variables were cross-analyzed to determine the threshold values on the basis of occupants' responses. Hirning et al. suggested that visual comfort (the border of perceptible disturbing) could be marked as DGP below 0.22. Meanwhile, Mangkuto et al. recommended DGP below 0.21. The participants indicated the presence of a glare sensation, even though most of the DGP values were under the border of imperceptible–perceptible at 0.35 [20]. The agreement, the tropical studies illustrated that Southeast Asians preferred a lower level of DGP in interior spaces. Additionally, according to those field studies, it can be noted that the experiments in a test room [20] and the filed surveys in occupied spaces may yield different results. Bian et al. [14] performed an empirical study in a test room in China using HDR images with a questionnaire and found that Chinese people were more durable to the recommended range of DGP. Karlsen et al. [32] studied the DGP with human test subjects in Denmark, finding that they were more tolerant to low illuminance levels and more sensitive to high illuminance levels than the DGP model would predict. Thus, a post-occupancy evaluation is a required method to thoroughly obtain occupants' feedback on daylight glare.

The preference of low DGP level was found in Southeast Asians; it can be inferred that they may prefer the low DGPs level since both share the same recommended range, and it is necessary to determine the applicable criteria for Thailand. For the purpose of this study, therefore, the DGPs model was selected for application with the field survey to evaluate daylight glare in occupied spaces; it was then analyzed with the occupants' subjective responses to develop its applicability to fulfill local occupants' preference. The finding from the study will be useful as an indicator of daylight glare in the building design stage, and as a variable incorporated into building control strategies. Moreover, in the context of tropical regions, there remains a lack of quantitative evidence-based research on the implementation of DGPs. The findings from this study are a remarkable statement to support further studies in Southeast Asia, and may suggest a possibility to recommend applicable criteria for building assessment when discussions on daylight glare are taken.

### 1.3. Aim and Objective

With reference to the above-mentioned introduction, this study aims to determine the applicable threshold values of DGPs on the basis of the responses from local occupants in Thailand. An acceptability-based approach was performed to evaluate the occupants' comfort using a questionnaire survey. Meanwhile, the DGPs index was calculated from the illuminance-based data in terms of vertical illuminance alone (Equation (2)), without considerations on the luminance of the light sources and the luminance contrast in one's field of view. The questionnaires' answers with their corresponding DGPs were paired together to indicate the subjective responses to DGPs. They were then analyzed by the interpretation of statistical approaches; their correlations were highlighted to determine the threshold values of each glare sensation level, which can be marked as references for Thailand.

The rest of this manuscript is organized as follows: Section 2 contains the methodological approaches under the post-occupancy stage of the buildings. Data collections including illuminance-based measurement and questionnaire survey are mentioned. The results and discussions are presented in Section 3. Sections 3.1 and 3.2 present the overview of surveyed data, while the threshold values of DGPs using statistical methods are presented in Section 3.3. Finally, Section 4 presents a conclusion with applicable DGPs criteria for Thailand.

## 2. Research Methods

### 2.1. Brief Introduction of the Target Office Spaces

#### 2.1.1. Target Buildings

For this study, we selected 8 high-rise office buildings with an operating air-conditioning (named office A-H) that are located in Bangkok (13°44′12.1812′′ N and 100°31′23.4696′′ E). All office spaces reflected the generic physical environment of an open-plan working station commonly found in the urban context of Thailand. Their profiles and basic information are summarized in Table 2. On-site data collection was performed for 3–5 days during working hours from 8:00 a.m. to 5:00 p.m. local time (UTC +7). This study focused on occupants working inside an open-plan space that served the largest number of occupants. Private office spaces or function rooms such as meeting rooms, conference rooms, service facilities, and recreation spaces were not included.

**Table 2.** Basic information of the surveyed case study office spaces.

| Subject | A | B | C | D | E | F | G | H |
|---|---|---|---|---|---|---|---|---|
| Office types | private | private | rental | private | private | private | rental | private |
| Surveyed period | APR 18 | JUN 18 | SEP 18 | SEP 18 | MAR 19 | MAR 19 | MAY 19 | SEP 19 |
| Construction year | 2005 | 2009 | 2015 | 2014 | 2011 | 2004 | 2015 | 2017 |
| Opening year | 2008 | 2010 | 2017 | 2016 | 2015 | 2008 | 2017 | 2019 |
| Building height (m) | 188 | 87 | 152 | 88 | 128 | 170 | 152 | 114 |
| Number of floors | 40 | 21 | 25 | 22 | 27 | 48 | 25 | 23 |
| Gross area (m$^2$) | 86,028 | 41,500 | 56,000 | 42,045 | 64,558 | 52,851 | 56,000 | 43,144 |
| Target floor | 14 | 6 | 11 | 17 | 17 | 32 | 15 | 11 |
| Floor area (m$^2$) | 3250 | 1826 | 1400 | 1320 | 1336 | 1257 | 1400 | 1274 |
| Target area (m$^2$) | 2755 | 1354 | 1050 | 971 | 925 | 912 | 1050 | 975 |
| Floor to ceiling (m) | 2.90 | 2.60 | 3.20 | 2.90 | 3.00 | 2.60 | 3.20 | 2.90 |
| Floor to floor (m) | 3.80 | 3.70 | 4.20 | 4.00 | 4.00 | 3.40 | 4.20 | 3.30 |
| Core position | central | eccentric | central | central | central | eccentric | central | eccentric |
| WWR | 0.61 | 0.72 | 0.98 | 0.72 | 0.55 | 0.73 | 0.98 | 0.70 |
| Glass types | laminated | IGU | IGU | IGU | IGU | laminated | IGU | IGU |
| Shading types | roller | venetian | roller | roller | venetian | roller | roller | venetian |
| SHGC (glass) | 0.48 | 0.34 | 0.34 | 0.32 | 0.28 | 0.46 | 0.34 | 0.35 |
| VLT (glass) | 0.74 | 0.86 | 0.86 | 0.79 | 0.55 | 0.68 | 0.86 | 0.77 |
| VLT (shading) | 0.03 | 0.10 | 0.15 | 0.24 | 0.07 | 0.02 | 0.15 | 0.10 |
| Occlusion rate (%) | 86 | 71 | 88 | 72 | 73 | 89 | 88 | 78 |
| Number of samples | 248 | 177 | 175 | 118 | 238 | 115 | 191 | 226 |

### 2.1.2. Target Occupants

The target occupants were the employee seated at certain positions in the surveyed buildings. They performed different tasks such as reading from a paper or working on a computer screen on the basis of their normal working situations. Concerning seat positions, the seated direction of facing-to-window was prioritized and focused on in this study since the glare perception from this direction had a higher potential of causing visual discomfort to the eye than that in the direction of parallel-to-window [29]. The actual furniture layouts in occupied spaces included various directions to the buildings' façades. The seats at a 45° diagonal to the façades or less were considered to be ones in the direction of facing-to-window [29,41], which was suitable for the DGPs model since it was strongly affected by the vertical illuminance and glare caused mostly by the quantity of light hitting the eyes [31]. Furthermore, the occupants in the perimeter and interior zones had different perceptions of lighting environments since they experienced different conditions of transmitted solar effects [40,42]. Thus, the occupants' survey in this study were only the ones who were inside the perimeter zone, which was the area within 15 feet from the façades [43].

### 2.1.3. Target Times of Data Collection

In this study, data collection under post-occupancy stage of the buildings involved an on-site measurement and a questionnaire survey. The dataset was collected twice a day at 11:00 a.m. and at 3:00 p.m. since a typical office peak-day occupancy time is distributed between 10:00 a.m. to 11:00 a.m., and from 2:00 p.m. to 3:00 p.m. [44]. Furthermore, the sun is naturally at its highest intensity at 10:00 a.m. and at 2:00 p.m., in terms of the location of Bangkok, Thailand [45]. These peak scenarios and times were selected for observing their consequences on the occupants in these conditions. However, during the field surveys, some occupants started to work at 10:00 a.m. or later due to the flexible working hour policy of their organizations. The specified time of 11:00 a.m. and 3:00 p.m. had a higher

potential to achieve the feedback from them in the questionnaire. Thus, these target times were applied with both data collection processes.

## 2.2. Illuminance-Based Data Measurement

The DGPs was calculated from illuminance-based data referring to the vertical illuminance term in Equation (2), since the seated positions in this study were that in the direction of facing-to-window experiencing the quantity of light arriving the eyes. The illuminance-based data were collected by light meter. All the devices for horizontal and vertical measures were tested and validated for their accuracy before on-site installation. They were placed vertically on a tripod to measure illuminance level sourcing from glazed façades at the same time, direction, and height level to compare all measured values to one another. The invalid device with a different rate higher than 5% [46] from others was excluded and replaced by a capable device.

During the field surveys, the on-site measurement was performed on the basis of the occupants' seat position as per the set-up illustrated in Figure 1. The vertical illuminance was vertically measured by an illuminance meter (UVC light meter, type k/j thermometer, LX-200SD, Lutron Electronic Enterprise CO.LTD, Taipei, Taiwan) installed on the adjustable tripod, applied at the seated-occupants' eye level (approximately 1.20 m from the floor), facing the direction that was experienced by the occupants in that particular position. This measuring set-up was placed in the area nearby the occupants' chair to maintain their normal activities. Meanwhile, the horizontal illuminance at desktop level (approximately 0.75 m from the floor) was collected by the installation of data logger with illuminance sensor (illuminance UV recorder, TR-74, T&D Corporation, Matsumoto, Japan) in automatic recording mode with time intervals of 1 min; the data were then derived by the target times. All the measuring processes started to operate and collect data while the occupants were asked to answer the questionnaire at 11:00 a.m. and at 3:00 p.m. After finishing at one seat position, the set-up was moved to the next position to start data collection with others.

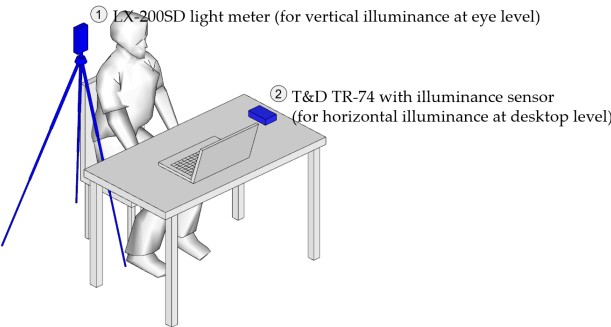

**Figure 1.** Simplified diagram of the measuring device installation for illuminance-based data.

During the post-occupancy evaluation, we kept all room elements or interior lighting systems operating in their normal state on the basis of the building management teams or occupants. The decision to let electric lighting be turned on was also taken. The study of Velds [47] found that the impact of artificial lighting contribution on users' perception was negligible. Pilot experiments in the study by Hirning et al. [48] suggested that turning the electric lighting off did not significantly influence the perception of a luminous environment. Even though electric lighting caused a bright scene, they were located above the line of sight and had a much smaller size compared to the daylight source. However, the controlled behavior of electric lighting and window operation, as well as the usage of shading devices were managed freely and depended on the occupants' interaction to maintain their normal working conditions during the field survey.

## 2.3. Subjective Evaluations of the Occupants

Subjective responses of the occupants were collected using a questionnaire survey. Paper questionnaires were collected from 1488 samples. The occupants were asked to answer or rate the level for the same set of questions at 11:00 a.m. and 3:00 p.m., which was the same period when on-site-measurement were told to collect physical data in a given seat position. As shown in Figure 2, the questions were divided into subjective and objective variables, and were structured as follows:

1. Time, date of survey, and identification number of the questionnaire.
2. Personal data (asking about gender/age/height/weight/type of work).
3. Occupants' comfort with the lighting environment (discomfort/slight discomfort/neutral/slight comfort/comfort).
4. Occupants' glare sensation vote (imperceptible/perceptible/disturbing/intolerable).
5. Occupants' sensation on lighting environment (extremely dim/very dim/dim/slight dim/neutral/slight bright/bright/very bright/extremely bright).

**Figure 2.** Paper-based questionnaire handed out to occupants.

The subjective variable measures included comfort and sensation votes. The comfort questions were rated on a 5-point semantic differential scale with endpoints of "discomfort" and "comfort". It was assumed that the scale was roughly linear, and ordinal values were assigned to each point along the scale as −2 (discomfort), −1 (slight discomfort), 1 (slight comfort), and 2 (comfort), with 0 as neutral at midpoint. Initially, the occupants indicated their comfort. They then took sensation questions with the glare sensation question about the source of comfort or discomfort by rating the level on a 4-point scale [16,20]. The predefined glare criteria were marked as 1 (imperceptible), 2 (perceptible), 3 (disturbing), and 4 (intolerable). Meanwhile, the lighting sensation was within a 9-point scale with endpoints of strong conditions such as −4 (extremely dark) and 4 (extremely bright), with 0 as neutral at midpoint. All occupants answered the questions by filling out a questionnaire form in the checkbox.

### 2.4. Classification of Sample Data and Statistical Analysis

For further analysis, as a single sample, each referred to the dataset of the occupant in a given seat position, containing both physical variables from an on-site measurement and subjective variables from a questionnaire survey, which were then paired together to indicate the occupants' responses to DGPs. For physical variables, we post-processed the measured illuminance-based data following Equation (2) to calculate the DGPs values. Meanwhile, for subjective variables, only the comfort and glare sensation votes were selected. Each was homogeneous in terms of the number of buildings and was individual in terms of its response feature.

To indicate the occupants' responses on DGPs, with respect to the answering scale, we assumed the comfort vote to be a nominal response by simplifying it to a nominal form. They were mentioned on the basis of their level as neutral occupants (equal to mid-scale at 0), discomfort occupants (below mid-scale at 0), and comfort occupants (above mid-scale at 0). Meanwhile, the glare sensation vote was assumed to be a binominal response by simplification to a binary form. Since the studies on discomfort glare determined the turnover point of glare perception by mid-scale at 2.5 [32,37], they were mentioned as glare-imperceived occupants (below mid-scale at 2.5) and glare-perceived occupants (above mid-scale at 2.5). Both were presented with their corresponding DGPs in terms of descriptive statistics to explain their general patterns.

For determination of the DGPs, both comfort and glare sensation votes were considered alongside DGPs. However, for statistical analysis, only samples with negative feedback were focused upon in order to determine the threshold values. Both were then classified as the class of "discomfort" by including the samples with forms of discomfort occupants and glare-perceived occupants, as mentioned above. Wienold [20] suggested four glare sensation levels, i.e., imperceptible, perceptible, disturbing, and intolerable. Each borderline was determined, including the threshold values obtained from the quartile calculation and predicted percentage of discomfort (PPD) methods, on the basis of Mangkutos' study [37]. For the quartile calculation method, we presented the lower quartile, median, and upper quartile values in the cumulative distribution histogram. Meanwhile, PPD was observed with the various discomfort percentages using a linear regression model, accounting for 10%, 25%, and 50%. From the two methods, those three values could be marked as the borders of imperceptible–perceptible, perceptible–disturbing, and disturbing–intolerable, respectively. The threshold values of these ranges were summarized, and their means were proposed as a reference for Thailand.

## 3. Results and Discussions

### 3.1. Surveyed Data

All occupants in the 1488 samples of this study were Thai employees. Among their working environments, the occupants worked inside daylit office spaces, located in an urban area of Bangkok, Thailand. Table 3 lists the profile and personnel data obtained from the questionnaire survey. It was observed that the majority (58.13%, i.e., 865 samples) of occupants were in their early thirties, with a median age of approximately 31 years. The distributions of gender were roughly equal, with a higher number of female occupants (58.87%, i.e., 876 samples). Most occupants (90.79%, i.e., 1351 samples) performed their work on the basis of a computer task. These characters were reflected in the demography of the occupants in this study.

The electric lighting was turned on as usually practiced throughout the day without any control from building management team or occupants. The personal task light could be found in some seat positions. However, it was not switched on during the survey period. Among these target buildings, the spaces were designed with a large glazing area (WWR), and with the high visible light transmittance (VLT) rate of the glazed façades compared to the buildings' façade recommendations of Thailand as 0.40 and 0.30–0.88, respectively [49]. As daylight penetrated into the interior spaces, however, the opening glass area was limited in this respect, since shading devices were lowered or closed, and it was kept in the down-position, as presented by the occlusion rate summarized in Table 2.

**Table 3.** Summary profile of the occupants as collected from the questionnaire survey.

| Survey Items | Answering | n | % |
|---|---|---|---|
| Age | 20–24-year-olds | 24 | 1.61 |
| | 25–29-year-olds | 407 | 27.35 |
| | 30–34-year-olds | 865 | 58.13 |
| | 35–39-year-olds | 108 | 7.26 |
| | 40–44-year-olds | 58 | 3.90 |
| | 45–50-year-olds | 13 | 0.87 |
| | 51–54-year-olds | 8 | 0.54 |
| | 55–60-year-olds | 5 | 0.34 |
| Gender | male occupants | 612 | 41.13 |
| | female occupants | 876 | 58.87 |
| Type of works | computer-based task | 1351 | 90.79 |
| | paper-based task | 137 | 9.21 |

For the DGPs model, its application enabled performance in the target spaces concerning the large-area sources of glare with no direct sun hitting the eyes, as per Wienolds' consideration [16,20]. Window size greatly influences occupants [50], with the daylight glare simulation study of Giovannini et al. [33] finding that the rate of VLT affected glare perception with the amount of vertical illuminance. However, shading device usage resulted in low conditions of light inside occupied spaces; daylight glare was never experienced by the users. Similar to what can be observed in this field survey, the high occlusion level of shading devices partially obstructed sunlight exposure. Thus, the direct sun or specular reflections had less effect on the occupants, which was suitable in terms of application of the DGPs model. Additionally, since the usage of shading devices was generally found, the measurements from the field survey should take into consideration the fact that the results in this study represented the shading situations or when occupants were unconsciously shaded.

For illuminance-based data, the horizontal illuminance at the desktop was derived from data logger at the target times. Meanwhile, the vertical illuminance at eye level was manually measured; it was then put into Equation (2) to calculate the DGPs values. The descriptive statistics of these physical variables are summarized in Table 4. In general, the horizontal illuminance was below the discomfort threshold of 2000 lux [25,26], ranging within 500–1000 lux (95.43%, i.e., 1420 samples) on the basis of the recommendations of Thailand [34]. Wienold [22] pointed out that horizontal illuminance could not take the spatial light distribution into account. The vertical illuminance might be favorable as an indication of discomfort glare for parameters of building control strategies. Focusing on the vertical illuminance, it can be observed with a wider range and larger SD. A possible explanation can be noted that its distribution varied with daylight conditions throughout the day and different seat positions [32,40]. Meanwhile, the horizontal illuminance at the desktop was maintained on the basis of the artificial light from the ceiling. The measured result was found to be below the discomfort threshold of 2670 lux [28], and mostly smaller than 1479 lux (95.43%, i.e., 1420 samples), which could be considered comfortable on the basis of the suggestion of Suk [29] for the seat position with facing-to-window direction.

On the basis of the illuminance-based data, we found that the occupants were likely to not be disturbed by daylight glare. The result of DGPs was also found smaller compared to the current reference. This implied that all DGPs values were smaller than 0.35. Wienold [20] suggested that DGPs below 0.35 could be referred to as imperceptible. Thus, from the comfort index evaluation, the occupants in this study were considered to be experiencing imperceptible glare. However, the occupants' responses to DGPs cannot be ignored since studies on daylight glare allowed for the differentiation between different levels of comfort classes, as proposed by Wienold [32,37], with it also addressing the

human variability to glare. The latter section presented these DGPs values with their corresponding subjective votes to indicate the occupants' responses to them.

**Table 4.** Summary of illuminance-based data and simplified daylight glare probability (DGPs).

| Surveyed Items | Max | Min | Mean | SD |
|---|---|---|---|---|
| Horizontal illuminance at the desktop level (lux) | 1755.01 | 542.88 | 764.20 | 105.29 |
| Vertical illuminance at the eye level (lux) | 2607.12 | 122.36 | 687.75 | 300.61 |
| DGPs | 0.34 | 0.19 | 0.23 | 0.02 |

*3.2. Correlation between DGPs and Its Subjective Variables*

Since the local occupants' preference must be carefully concerned, we investigated the comfort and glare sensation votes by pairing with their corresponding DGPs values. On the basis of a questionnaire with its interpretation from the answering scale mentioned in Section 2.4, for the comfort votes, we show in Table 5 that most occupants voted for neutral (30.65%, i.e., 456 samples). The vote presented more positive values for the comfort occupants (47.11%, i.e., 701 samples) rather than the discomfort occupants (22.24%, i.e., 331 samples), for which 13.17% (196 samples) referred to it as a discomfort, and 9.01% (135 samples) as a slight discomfort. After occupants answered the comfort questions, they were asked to rate the glare sensation votes. Table 6 shows that most of them were glare-imperceived occupants (65.72%, i.e., 972 samples) with some glare-perceived occupants (34.68%, i.e., 516 samples). Of the occupants who indicated the presence of glare, 32.12% (478 samples) responded to it as disturbing, and 2.55% (38 samples) as intolerable. From the field survey, all DGPs values were revealed to be below 0.35, which could be denoted as imperceptible glare. All the occupants in this study were supposed to achieve visual comfort without the presence of glare. However, a group of occupants with negative responses could be observed from the subjective questionnaire survey.

**Table 5.** Summary of the DGPs based on the level of comfort vote from the questionnaire survey.

| Comfort Vote | n | % | Max | Min | Mean | SD |
|---|---|---|---|---|---|---|
| Discomfort | 196 | 13.17 | 0.34 | 0.21 | 0.25 | 0.02 |
| Slight discomfort | 135 | 9.07 | 0.34 | 0.20 | 0.24 | 0.02 |
| Neutral | 456 | 30.65 | 0.28 | 0.19 | 0.23 | 0.02 |
| Slight comfort | 372 | 25.00 | 0.26 | 0.19 | 0.22 | 0.01 |
| Comfort | 329 | 22.11 | 0.25 | 0.19 | 0.21 | 0.01 |

**Table 6.** Summary of the DGPs based on the level of glare sensation vote from the questionnaire survey.

| Glare Sensation Vote | n | % | Max | Min | Mean | SD |
|---|---|---|---|---|---|---|
| Imperceptible | 117 | 7.86 | 0.24 | 0.19 | 0.21 | 0.01 |
| Perceptible | 855 | 57.46 | 0.28 | 0.19 | 0.22 | 0.01 |
| Disturbing | 478 | 32.12 | 0.34 | 0.19 | 0.24 | 0.02 |
| Intolerable | 38 | 2.55 | 0.34 | 0.22 | 0.27 | 0.03 |

Focusing on the occupants with negative feedback, it might seem to be a contradiction that they responded to glare under the environment with low vertical illuminance. From Table 5, the mean values of DGPs as slight discomfort and discomfort were 0.24 and 0.25, respectively. Meanwhile, from Table 6, negative response to glare sensation as disturbing and intolerable were 0.24 and 0.27, respectively, which could be seen to be small compared to Wienolds' study [20]. In his study, the results from human test subjects were marked as imperceptible, perceptible, disturbing, and intolerable at 0.33,

0.38, 0.42, and 0.53, respectively. However, according to the agreement in tropical regions, the results in this study were similar to the field studies on discomfort glare in Malaysia [40] and Indonesia [37]. Both revealed that most of the DGP and DGPs values were below 0.35, and a group of discomfort occupants could be found therein. Moreover, the experiments in a test room might yield different results from the field survey under the contextual study [37]. This was possibly attributed to daylight obstruction since the shading devices were lowered. Giovannini et al. [33] and Sadeghi et al. [51] found that human interactions with shading devices could lessen DGPs. Under the low lighting environments, it is important to note that contrast-based glare might be a considerable concern [32]. However, the limitations of illuminance-based study can never account for this, unless the contrast itself contributes to a significant increase in the vertical illuminance [31,32], and it should be also noted that all responses of glare in this study were reported under the shading situations.

With respect to the index of visual comfort probability (VCP) introduced by Guth [52] to evaluate the percentage of the observers' population who would be considered comfortable in a given lighting environment, we found that the suggestion is strictly higher than 70% to ensure occupants achieve visual comfort. On the basis of the percentage stacked bar of the occupants' responses with their corresponding DGPs, we show in Figure 3 that the turnover point of the occupants to start to report a discomfort vote (comfort vote below mid-scale at 0) higher than 30% (49.32%) was at a DGPs of 0.23, and a glare-perceived vote (glare sensation vote above mid-scale at 2.5) higher than 30% (38.47%) was at a DGPs of 0.22, which can be seen in Figure 4. Both negative percentages were likely to rise gradually when the DGPs increased, even though they were under the border of imperceptible–perceptible at 0.35 [20]. These results indicated the preference on lower DGPs level; its references need to be modified to fulfill the needs of local occupants. It is therefore worthwhile to further investigate subjective responses to thoroughly determine DGPs threshold values.

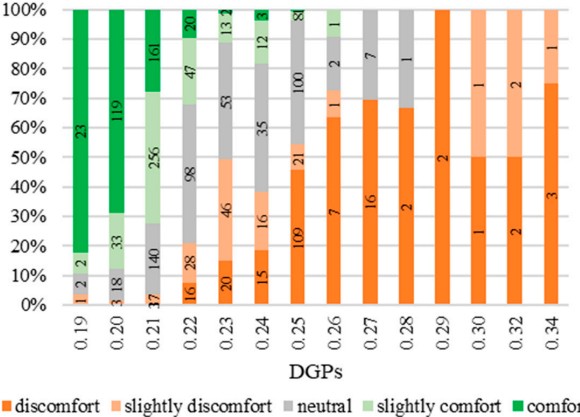

**Figure 3.** Percentage stacked bar of the comfort votes with each value of DGPs.

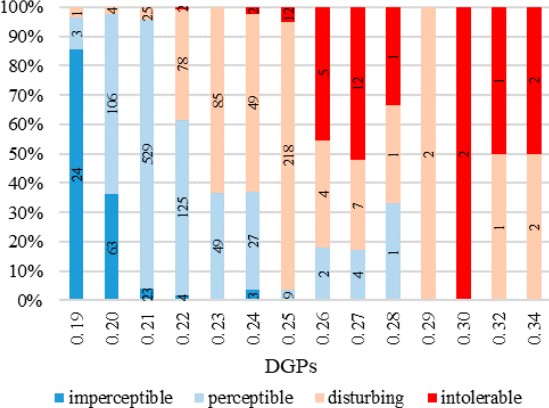

**Figure 4.** Percentage stacked bar of glare sensation votes with each value of DGPs.

Concerning the usability of the DGPs model, we found it to play a significant role in prediction of glare perception. The DGPs values with their corresponding subjective votes, i.e., comfort and glare sensation votes, were plotted together using bubble plots mapped with their linear regression lines in Figures 5 and 6, respectively. Both resulted in a similar trend and showed a reasonable correlation, with a larger coefficient of determination ($R^2$ = 0.548) in glare sensation vote model. Thus, the accuracy of DGPs can be considered as a method for daylight glare evaluation with respect to the occupants' responses. Additionally, it can be seen that both of their regression lines revealed a comparable turnover point of DGPs on the basis of their subjective mid-scales. From the comfort model, the neutrality (mid-scale of 0) can be read as DGPs of 0.24. Meanwhile, the glare sensation model revealed a midpoint (mid-scale of 2.5) as DGPs of 0.23. These results indicated that the occupants' responses started to shift inversely by the DGPs border as 0.23–0.24. Nonetheless, as seen in Figure 6, the threshold values of imperceptible–perceptible (scale at 1.5), perceptible–disturbing (scale at 2.5), and disturbing–intolerable (scale at 3.5) can be read as 0.20, 0.23, and 0.27, respectively. It can be deduced that the local occupants prefer a lower DGPs level than that of current references. This evidence led to the finding that the applicable criteria under the contextual study in Thailand were distinctly different from the suggested values. For further analysis in the latter section, we highlighted the negative feedback of occupants denoted in terms of the class of discomfort in order to determine the threshold values of DGPs.

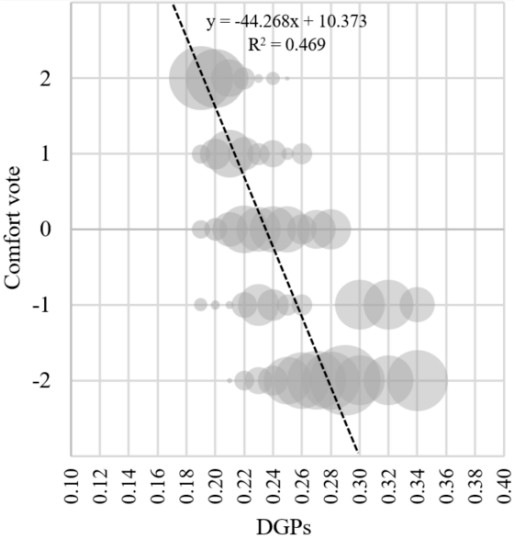

**Figure 5.** Correlation between the comfort votes with their corresponding DGPs.

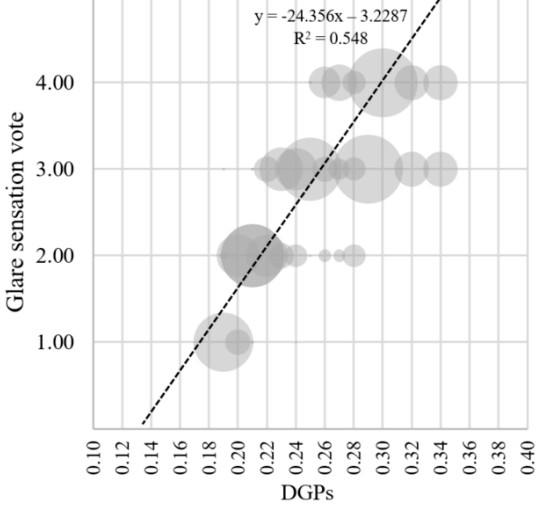

**Figure 6.** Correlation between the glare sensation votes with their corresponding DGPs.

### 3.3. Simplified Daylight Glare Probability (DGPs) Threshold Values

To determine the DGPs threshold values, we proposed the four glare sensation levels through considering three thresholds as imperceptible–perceptible, perceptible–disturbing, and disturbing–intolerable. Statistical analysis was applied to the dataset to determine each threshold value in accordance with the subjective votes from occupants. Since comfort and glare sensation votes revealed the same trend as mentioned above, both were included to be discomfort class, as mentioned in Section 2.4. However, the number of glare-perceived occupants was found to be higher than that of discomfort occupants. Some might express the presence of glare sensation with an acceptable comfort vote. Therefore, only discomfort occupants with negative responses to glare sensation votes were selected. From a total 331 discomfort occupants and 516 glare-perceived occupants, only 313 were eligible to be referred as discomfort class, as summarized in Table 7. On the basis of the statistical *t*-test with a 5% significance level, we found the DGPs to be the index that could represent the occupants' responses most effectively (*p*-value = 0.018). Its mean value was 0.25, which was comparable to the study of Mangkuto et al. [37] that defined the class of discomfort occupants in Indonesia as DGPs of 0.26. According to the agreement in tropical studies, glare perception of Thai occupants was substantially lower than the current references [20], allowing a possibility to lower the threshold value for each glare sensation level. Wienold proposed the thresholds of DGPs in his simulation study [20] with reference to the descriptive statistics of human test subjects' assessment from the test room study. For this study, we determined the threshold values by applying the dataset of discomfort class to the statistical approaches, i.e., the quartile calculation and PPD methods, since they were proposed on the basis of the local occupants' responses in Mangkutos' study.

**Table 7.** Summary of the DGPs based on the occupants' class of discomfort.

| Class | n | % | Max | Min | Mean | SD | t Critical | t Calculated | *p*-Value |
|---|---|---|---|---|---|---|---|---|---|
| Discomfort | 313 | 21.03 | 0.34 | 0.21 | 0.25 | 0.02 | 1.48 | 1.95 | 0.018 |

Histograms showing the cumulative percentage of the discomfort class under various DGPs are presented as the curved line in Figure 7. The lower quartile, median, and upper quartile values can be determined using 25%, 50% and 75% of the cumulative percentages on the *y*-axis, respectively. The correlating values can be read with the histogram curve projection on the *x*-axis. The threshold values were simultaneously determined by the PPD method, referring to the data set in Figures 3 and 4. The linear regression model of PPD presented in scatter plots, as shown in Figure 8. It can be seen that the DGPs had a strong correlation with the subjective votes. The finding thresholds can be expected to be responsive to the occupants' responses. According to Mangkutos' study [37], the PPD of 10% may be considered to be at the imperceptible–perceptible threshold, taking an analogy with PPD in the context of thermal comfort, for which values lower than 10% are suggested to correlate with a neutral sensation, providing the most acceptable range [9]. The 50% value gives a statistical threshold above which a majority of subjects feel discomfort. Therefore, this is suggested as the disturbing–intolerable threshold. The 25% value refers to the border in the middle ranges, i.e., the perceptible–disturbing threshold. Moreover, the 25% value is highlighted as the border of discomfort occupants for the VCP index. The study of Carlucci et al. [36] on the comparison of visual comfort indices modified some numerical values to meet the semantic descriptions with a comfort scale resolution; the acceptable rate of VCP was possibly increased from 70 to 75 with unacceptance at 25–30%, which agrees with the consideration of 25% in this study.

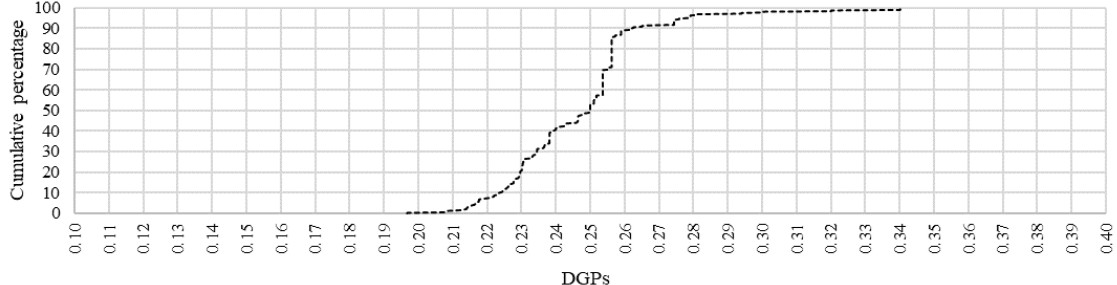

**Figure 7.** Cumulative histograms showing the percentage of discomfort class with various values of DGPs.

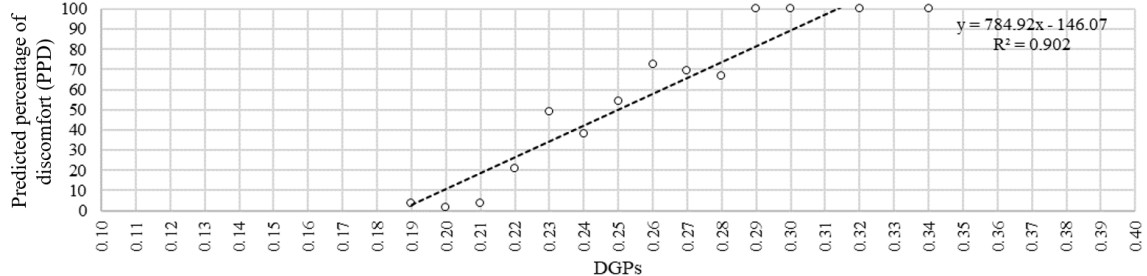

**Figure 8.** Predicted percentage of discomfort class with various values of DGPs.

Table 8 summarizes the threshold values on the basis of those methods, with their agreement being observable. The DGPs values that correspond to the lower quartile, the median, and the upper quartile are comparable to those from 10%, 25%, and 50% of PPD, respectively, with negligible differences. These values are suggested as the threshold of imperceptible–perceptible: DGPs = 0.20–0.23, perceptible– disturbing: DGPs = 0.22–0.25, and disturbing–intolerable: DGPs = 0.25–0.26. To summarize DGPs range with reference to their mean values as borderline, we can mark the suggestions for this study as imperceptible: DGPs = below 0.22, perceptible: DGPs = 0.22–0.24, disturbing: DGPs = 0.24–0.26, and intolerable: DGPs = above 0.26. For the intolerable condition, this finding agrees with previous studies on DGP in Malaysia [40] and Indonesia [37], which determined the threshold values of discomfort and intolerable glare to be below 0.26 and 0.24–0.26, respectively. With the agreement in tropical studies, it can be concluded that Thai occupants prefer a lower DGPs level since the field survey revealed their sensitivity to daylight glare, giving a possible explanation that several samples with DGPs below 0.35 are responded to with negative feedback. Additionally, the threshold of perceptible–disturbing summarized in Table 8 is comparable to the results in Figures 5 and 6, which present DGPs on the basis of the mid-scale of subjective vote as 0.23–0.24. This can be recognized as the primary border for the turnover point between comfort–discomfort [32]. Thus, the median quartile and the PPD of 25% are indicators that can distinctly classify the occupants' responses, with this threshold at the level of perceptible–disturbing requiring careful concern in terms of the daylight glare evaluation.

**Table 8.** Suggested thresholds of DGPs at three levels of glare sensation based on the quartile calculation method and predicted percentage of discomfort (PPD) method.

| Thresholds | Quartile Method | PPD Method | Mean | Difference (%) |
|---|---|---|---|---|
| Imperceptible–perceptible | 0.23 | 0.20 | 0.22 | 13.04 |
| Perceptible–disturbing | 0.25 | 0.22 | 0.24 | 12.00 |
| Disturbing–intolerable | 0.26 | 0.25 | 0.26 | 3.85 |

Regarding the preference for lower DGPs, there is a fair interpretation that Thai occupants would prefer a lower level of daylight entering interior spaces to avoid glare falling directly in the eyes. The study on light source type with visual preference from French office workers [53] found that the preferred illuminance level on the desk under daylight was lower than those under electric light when the participants were allowed to choose their own visual environment. Several studies in tropical regions suggested that the amount of preferable or acceptable daylight for interior spaces was likely to be lower than the recommendations in illumination standards. The study by Mangkuto et al. on daylight perception in a library [37] and classrooms [54] in Indonesia suggested that lower threshold values of daylight factor, recommended illuminance level, and DGP were preferred in comparison to those of the current national standards. The study by Dahlan et al. [55] on daylight acceptance in residential buildings in Malaysia suggested that the occupants preferred a low level of daylight.

In Thailand, most studies focus on daylight utilization approaches, such as work plane illuminance [56,57], daylight factor [58], and light shelf [39,58], without any concerns for the occupants' perspective. There are a few studies mentioning the users' expectations on low lighting environments. The field study on recommended light level in office spaces in Thailand by Ramasoot et al. [57] found that the comfortable work plane illuminance level based on occupants' opinion was around 300–400 lux. Meanwhile, the local standards suggested it to be above 500 lux [34], which was too high to fulfill them and led to unnecessary energy use. The field study of Chaloeytoy et al. [42] on acceptable interior lighting in office spaces in Thailand and Singapore found that the occupants usually lowered the internal shading devices to avoid discomfort glare and reduce over-brightness. Without shading devices usage, DGP and horizontal illuminance at the desktop level could be beyond the occupants' acceptance. Nevertheless, in terms of non-visual factors, the study of Puchongprawet [59] on white skin obsession in Thailand noted that the "white skin preference" phenomenon made Thais avoid direct sunlight exposure to not get tanned by the sun's burning rays. Moreover, since the amount of daylight in Thailand is relatively high, it is important to carefully provide indoor conditions. Thai residential buildings are practically designed and built with several external shading devices to avoid sunlight penetration [45]. All interior spaces are dominated by shadow covering, and therefore Thais are accustomed to indoor living with shading environments [60]. As daylight is rarely utilized in interior spaces, the local preference on low daylighting conditions leads to a possibility to lower the reference values of daylight glare criteria to achieve visual comfort. The suggested DGPs in this study can be considered as a notable statement for improvement in the understanding of occupants' perspective.

Daylight correlated moderately with beliefs about the importance of lighting. Humans can obtain many benefits in terms of well-being, comfort, and productive work environments for building occupants [2,3,37,61]. The study of Well [62] on occupants' attitudes toward windows found that daylight is preferred and that it was better for the occupants' eyes to work by daylight than by electric lighting. However, the preference of low daylight glare was revealed by the occupants in this study. High levels were generally viewed as more unpleasant than lower levels, which suggested a strong psychological link to glare and overheating [61]. To utilize the considerable levels, the controlled strategies for approaching daylight and avoiding discomfort glare must be carefully considered to fulfill the occupants in terms of considerations on seat positions [29], window characteristics [61,62], and shading device controls [63], alongside the DGPs thresholds found in this study. Additionally, it is important to note that this study only focused on the daylight glare souring from glazed façades. Visual environments in daylit office spaces are combined systems supplied by daylight and electric lighting [61]. Well [62] said that the occupants' assessment of daylight levels overestimated the proportion of daylight that they worked under with their distance from the windows; some might be misled by the electric lighting. A post-occupancy evaluation by Christoffersen et al. [64] in Danish office buildings showed that the occupants' preference on daylight for workplaces located near windows should be concerned with electric lighting even when there was sufficient daylight. Thus, the studies on other lighting metrics or discomfort glare indices are further required to examine occupants' preferences concerning actual physical conditions with different glare sources.

Since daylight glare evaluation found in Thai illumination standards was opaque, the DGPs model has potential for the next steps in addressing and determining as the applicable daylight glare criteria. Further, on the basis of the discrepancy between the preferred DGPs and the given references, the local occupants required different threshold values from those in recent studies [20]. To enhance visual comfort, it is necessary to consider this and to carefully integrate local requirements by setting up the DGPs threshold values in this study to be a renewed reference, which can be recognized as an initial step to pave the way for an improvement of national standards. The guidelines for interior lighting design [34] and green government office design guidelines for new construction [35], which currently encourage building designers or occupants to utilize daylight without glare control strategies, need to be concerned with occupants' responses. In more practical terms, the suggested thresholds in this study can be developed as an indicator of daylight glare evaluation in early design and in improving building performance under the post-occupancy stage when decisions regarding glare perception are taken.

## 4. Conclusions

To enhance occupants' comfort in glazed façade buildings in terms of experiencing quantity of light arriving to the eyes, we performed a field survey in daylit office spaces in Thailand to determine daylight glare criteria on the basis of the local occupants' responses. The DGPs model was explored under post-occupancy evaluation involving an on-site measurement and a questionnaire survey. The results showed the potential of DGPs as an indicator for prediction of daylight glare. However, the occupants' preference for a lower level than that found in current references was found. Statistical analyses were employed to determine glare sensation level on the basis of the occupants' responses; the following are suggested as threshold values for imperceptible–perceptible: DGPs = 0.22; perceptible–disturbing: DGPs = 0.24; and disturbing–intolerable: DGPs = 0.26. The agreement from this study with other tropical studies [37,40] helps conclude that Thai occupants prefer a low level of daylight glare. This can be referred to as quantitative evidence to support further study on daylight glare in tropical regions, allowing for differentiation between glare levels as proposed by Wienold [20].

The findings lead to an improvement in understanding of the perspective of local occupants. There is a possibility to modify current daylight glare criteria by addressing the DGPs model with its suggestions found in this study. Moreover, Thailand illumination standards [34,35], which currently focus on daylight utilization without any specific glare controls, should be revised further in accordance with the local requirements. With an appropriately controlled strategy, the buildings' performance can be potentially developed to provide appropriate conditions into interior spaces, which can practically be expected to take a step towards improving occupants' comfort.

## 5. Limitations

The sky conditions and outdoor daylight illuminance play an important role in daylight utilization along with interior lighting environments. However, as the acceptability-based approach in this study was only focused on the occupants' perspective with a limit of environmental factor considerations, we must consider the contextual conditions in a further study in order to provide an explanation for physical variables or occupants' feedback. More detailed analysis is needed in terms of lowlight-dominant environments or shading situations. More evidence from research on other glare indices or glare sources is also needed and should be further incorporated to ensure local preference. Moreover, the findings are only applicable for occupants' view in the direction of facing-to-window, and the effect of building orientations is not mentioned since it could not be controlled under the filed survey in actual buildings. Further study is required to pursue other possibilities for a variety of seat positions.

**Author Contributions:** Conceptualization, K.C. and M.I.; methodology, K.C. and M.I.; formal analysis, K.C.; investigation, K.C. and M.I.; writing—original draft preparation, K.C.; writing—review and editing, K.C., M.I., and S.-C.C.; supervision, M.I. and S.-C.C. All authors have read and agreed to the published version of the manuscript.

**Funding:** Tokyo Metropolitan Government Platform collaborative research grant.

**Acknowledgments:** This research was conducted by Tokyo Metropolitan Government Platform collaborative research grant (representative: Masayuki Ichinose). All of the support is gratefully acknowledged.

**Conflicts of Interest:** The authors declare no conflict of interest.

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
