# Peer review of "Determination of the Simplified Daylight Glare Probability (DGPs) Criteria for Daylit Office Spaces in Thailand"

_buildings, doi:10.3390/buildings10100180_

Round 1

Reviewer 1 Report

The manuscript compares the simplified DGP metrics (DGPs) based on vertical illumination level with subjective evaluation of glare in a set of office rooms in Thailand. The windows were shielded in all investigated rooms, so DGPs metrics should be appropriate (full DGP metrics reflects also the contrast between the source of glare and the background). Based on the answers of employees, authors found DGPs threshold for disturbing and intolerable glare. Beside the proposed DGPs thresholds, the main finding is that Thai people like less sunlight in office rooms than internationally recommended levels.

As mentioned by authors, acceptable levels of glare can depend on cultural traditions and on environmental characteristics of the office rooms  (air temperature and humidity, orientation of the room according cardinal points, ...). The authors plan to examine these parameters in more depth in future work.

The manuscript is written very clearly and all conclusions are supported by experimental data. The research method and data processing are described in detail. I found only minor problems that can be easily corrected:

  1. Equation 1: index "i" that is currently placed before the fraction, has to be placed under the symbol of sum.
  2. Plot type in Figures 4 and 5 is confusing - the fitting lines go apparently out of the main group of data points. The reason of this is simple: the circles in plots are overlapping and (for example) "comfort votes 2" are in Figure 4 practically evenly distributed between 0.19 and 0.25. In fact, big majority of points lie between 0.19 and 0.21, as can be seen in Figure 1. In my opinion, another type of plot would better present the data. I can recommend "bubble plot", for example, with responses grouped by DGPs (step 0.2 or similar). 

Author Response

Respond to reviewer 1 comments

Dear Reviewers 1,

Thank you for your considerations. It was with great pleasure to receive your comments. I greatly appreciate your feedback and take it to be considered for the revised-version. The manuscript intent remains as determination of applicable daylight glare criteria for Thailand. Revision details (as blue letter) are listed following your suggestions as:

The manuscript compares the simplified DGP metrics (DGPs) based on vertical illumination level with subjective evaluation of glare in a set of office rooms in Thailand. The windows were shielded in all investigated rooms, so DGPs metrics should be appropriate (full DGP metrics reflects also the contrast between the source of glare and the background). Based on the answers of employees, authors found DGPs threshold for disturbing and intolerable glare. Beside the proposed DGPs thresholds, the main finding is that Thai people like less sunlight in office rooms than internationally recommended levels.

As mentioned by authors, acceptable levels of glare can depend on cultural traditions and on environmental characteristics of the office rooms  (air temperature and humidity, orientation of the room according cardinal points, ...). The authors plan to examine these parameters in more depth in future work.

The manuscript is written very clearly and all conclusions are supported by experimental data. The research method and data processing are described in detail. I found only minor problems that can be easily corrected:

1.Equation 1: index "i" that is currently placed before the fraction, has to be placed under the symbol of sum.

Respond to reviewer 1: I apologize for this unconscious mistake, the index "i" was moved to place under the symbol of sum (line 72), please find it in the revised manuscript.

2.Plot type in Figures 4 and 5 is confusing - the fitting lines go apparently out of the main group of data points. The reason of this is simple: the circles in plots are overlapping and (for example) "comfort votes 2" are in Figure 4 practically evenly distributed between 0.19 and 0.25. In fact, big majority of points lie between 0.19 and 0.21, as can be seen in Figure 1. In my opinion, another type of plot would better present the data. I can recommend "bubble plot", for example, with responses grouped by DGPs (step 0.2 or similar). 

Respond to reviewer 1: To indicate the occupants’ responses on DGPs, the format of bubble plot was replaced for the data sets in Figure 5 (changed from Figures 4.) and Figure 6 (changed from Figures 5.) to show their membership size in each DGPs value. Their linear regression model were mapped alongside to reveal the usability of DGPs and occupants’ responses, and to assume the border between each glare sensation level aiming at the possibilities to lower threshold value for each in the latter section,  please find it in the revised manuscript.

If you need further information, please do not hesitate to contact me.
Your feedback will be highly appreciated. Thank you very much.

Reviewer 2 Report

In this work, an interesting discussion on daylight probability criteria for office spaces has been carried out. In my opinion, the manuscript is a quite good contribution to the large field of people's comfort in interior spaces. The presented study aims to determine daylight glare thresholds based on the occupants’ responses. The daylight glare evaluation was performed with a simplified probability model (DGP) in different office buildings; moreover, on-site measurements, and questionnaire surveys were collected. The article is well organized and the authors clearly describe the objectives and methodology of their work.

In my opinion, the paper can be published provided that the following modifications are performed:

- the background material presented in the paragraph “Introduction” should be expanded adding some reference for other building destinations. The authors mention only office building, but the same problem is presented in residential buildings and hospitals as well. As for this, one suggestion is to add to the list of references the following papers:

https://doi.org/10.1016/j.enbuild.2006.03.001

https://doi.org/10.3390/en13082116

https://doi.org/10.1016/j.egypro.2018.08.027

I also suggest adding a table with the acronyms used in the paper.

Author Response

Respond to reviewer 2 comments

Dear Reviewers 2,

Thank you for your considerations. It was with great pleasure to receive your comments. I greatly appreciate your feedback and take it to be considered for the revised-version. The manuscript intent remains as determination of applicable daylight glare criteria for Thailand. Revision details (as blue letter) are listed following your suggestions as:

In this work, an interesting discussion on daylight probability criteria for office spaces has been carried out. In my opinion, the manuscript is a quite good contribution to the large field of people's comfort in interior spaces. The presented study aims to determine daylight glare thresholds based on the occupants’ responses. The daylight glare evaluation was performed with a simplified probability model (DGP) in different office buildings; moreover, on-site measurements, and questionnaire surveys were collected. The article is well organized and the authors clearly describe the objectives and methodology of their work.

In my opinion, the paper can be published provided that the following modifications are performed:
- the background material presented in the paragraph “Introduction” should be expanded adding some reference for other building destinations. The authors mention only office building, but the same problem is presented in residential buildings and hospitals as well. As for this, one suggestion is to add to the list of references the following papers:
https://doi.org/10.1016/j.enbuild.2006.03.001
https://doi.org/10.3390/en13082116
https://doi.org/10.1016/j.egypro.2018.08.027

Respond to reviewer 2: The introduction tried to highlight current situations in working environments, and paved the way for aiming at the study on office spaces in latter section. However, as for your suggestions, background in all building types was introduced in line 29 – 34 [1, 2] with discussion on daylit offices in line 519 - 538 [3] were added to support the body of knowledge in occupants’ preferences with visual environments, please find it in the revised manuscript.

I also suggest adding a table with the acronyms used in the paper.

Respond to reviewer 2: The acronyms list was added in the end of introduction part (line 60), please find it in the revised manuscript.

If you need further information, please do not hesitate to contact me.
Your feedback will be highly appreciated. Thank you very much.

------------------------------------------------------------------------------------------------------------------------------------------------------

[1]   Cesari, S.; Valdiserri, P.; Coccagna, M.; Mazzacane, S. The energy saving potential of wide windows in hospital patient rooms, optimizing the type of glazing and lighting control strategy under different climatic conditions. Energies. 2020, 13, 2116. [https://doi.org/10.3390/en13082116]

[2]    Cesari, S.; Valdiserri, P.; Coccagna, M.; Mazzacane, S. Energy savings in hospital patient rooms: the role of windows size and glazing properties. Energy Procedia. 2018, 148, 1151-1158. [https://doi.org/10.1016/j.egypro.2018.08.027]

[3]    Galasiu, A.D.; Veitch J.A. Occupant preferences and satisfaction with the luminous environment and control systems in daylit offices: a literature review. Energy and Build. 2006, 38, 728-742. [https://doi.org/10.1016/j.enbuild.2006.03.001]

Reviewer 3 Report

The paper faces the evaluation of the simplified daylight glare probability based on the responses of occupants of eight open-plan office spaces located in Bangkok. In general the paper is clear, but some major issues should be addressed before the paper could be considered for publication.

Major revision

  • Introduction: the PWV as well as any other index not used in the paper should be avoided, while used indices, as PPD, should the mentioned;
  • Section 1.1: in the line 74 the authors write “…Glare perception could be simply detected by horizontal illuminance on the desktop level…”, but in reference [15] is reported that “….Desktop illuminance was among the weakest of the existing lighting design metrics ….”. A research paper should highlight these differences;
  • Section 1.1, line 74:, further references about the useful daylight illuminance values on a work plane could be useful for readers to understand the complexity of the topic;
  • Section 2.2: the accuracy of the used illuminance meters should be declared;
  • Section 2.2, line 228-231: references 42 and 43 indicate that the impact of artificial lighting contribution on users’ perception was negligible. In authors opinion, does that also apply to glare?
  • Section 2.2, line 228-231: even though the impact of artificial lighting contribution on users’ perception was negligible, this is not true for illumination values. Please, better justify your research hypothesis;
  • Figure 1: photos, rather than a diagram, of the measurement set up as well as the luminous environment could be more useful for readers;
  • Section 2.3: please insert an extract of the questionnaire;
  • Table 4: please, explain how the authors calculate the values listed in the table;
  • Section 3.1, line 340: please, check the references.

Author Response

Respond to reviewer 3 comments

Dear Reviewers 3,

Thank you for your considerations. It was with great pleasure to receive your comments. I greatly appreciate your feedback and take it to be considered for the revised-version. The manuscript intent remains as determination of applicable daylight glare criteria for Thailand. Revision details (as blue letter) are listed following your suggestions as:

The paper faces the evaluation of the simplified daylight glare probability based on the responses of occupants of eight open-plan office spaces located in Bangkok. In general the paper is clear, but some major issues should be addressed before the paper could be considered for publication.

Major revision

*Introduction: the PWV as well as any other index not used in the paper should be avoided, while used indices, as PPD, should the mentioned;

Respond to reviewer 3: The PMV was mentioned to overview the occupants’ comfort evaluation in working environments. Since PPD was a product of PMV, it was additional mentioned (line 46) along the line to introduce concepts with consideration on discomfort percentage, which was considerably apply with the determination on DGPs threshold values.

*Section 1.1: in the line 74 the authors write “…Glare perception could be simply detected by horizontal illuminance on the desktop level…”, but in reference [15] is reported that “….Desktop illuminance was among the weakest of the existing lighting design metrics ….”. A research paper should highlight these differences;

Respond to reviewer 3: To clarify this message, the wording was changed from “…Glare perception could be simply detected by horizontal illuminance on the desktop level…” to “…Lighting design metrics  could be simply provided by horizontal illuminance on the desktop level due to its ease of use and prevalence in practice…” (line 84). This was done to briefly introduce the background of illuminance-based study. Then the main focus was shift into the vertical illuminance since it was more capable than the horizontal illuminance to predict glare perceptions.

*Section 1.1, line 74:, further references about the useful daylight illuminance values on a work plane could be useful for readers to understand the complexity of the topic;

Respond to reviewer 3: Further references about the useful daylight illuminance (UDI) [1, 2, 3] were provided (line 86 - 91) to overview a background on work plane illuminance; their threshold values were mentioned to be referred in the section of results and discussions.

*Section 2.2: the accuracy of the used illuminance meters should be declared;

Respond to reviewer 3: To validate the illuminance meters, all the measuring devices for horizontal (named as illuminance UV recorder, T&D TR-74) and vertical illuminance (named as UVC light meter, type k/j thermometer, LX-200SD) were tested before installations. They were placed vertically on tripod to measure illuminance level sourcing from glazed facades at the same time, direction, and height level to compare all measured values to one another. The invalid device with a different rate higher than 5% from others was excluded and replaced by the capable one. Please find this message in line 229 -234.

*Section 2.2, line 228-231: references 42 and 43 indicate that the impact of artificial lighting contribution on users’ perception was negligible. In authors opinion, does that also apply to glare?; Section 2.2, line 228-231: even though the impact of artificial lighting contribution on users’ perception was negligible, this is not true for illumination values. Please, better justify your research hypothesis;

Respond to reviewer 3:  Concerning the impact of artificial lighting, the visual environments in daylit office spaces were a combined systems supplied by daylight and electric lightings. A post-occupancy evaluation by Christoffersen et al. [4] in Danish office buildings showed that the occupants’ preference on daylight for workplaces located near windows should be concerned with electric lightings even when there was sufficient daylight. However, field investigation in this study only focused on the light source from glazed facades aiming at the occupants’ responses to daylight glare. Furthermore, in terms of the occupants’ survey, the study of Well [5] on occupants’ attitudes toward windows found that the occupants’ assessment of daylight levels were overestimate the proportion of daylight that they worked under proportionally with their distance from the windows; some might mislead from the electric lightings. Thus, even though electric lights were located above the line of sight and had a much smaller size compared to the daylight source. It could affect the occupants’ perception by falling into their eyes or creating a contrast-based glare [6]. The studies on other lighting metrics or discomfort glare indices are further required to examine and support occupants’ preference concerning actual physical conditions with different glare sources.

               To convey these message, the role of artificial light in field study was addressed in line 249 - 253 to scope the boundaries of this study. The assumption on a contrast-based glare was addressed in line 396 - 399 to mention other glare source which could be possibly found in occupied spaces. Additional discussion on artificial lighting was added in line 519 – 538, to state the finding in this study with an overview of occupants’ preferences on daylight and electric lightings, which allowed possibilities for further studies on other lighting metrics or discomfort glare indices to strengthen the occupants’ preference.

*Figure 1: photos, rather than a diagram, of the measurement set up as well as the luminous environment could be more useful for readers;

Respond to reviewer 3: The diagram represented the measurement set up in occupied spaces since all documents or photographs taken in the buildings were not allow to be published; they were strictly mentioned in the non-disclosure undertaking with the building stakeholder to keep them as confidential information. Thus, the buildings’ characteristics were mentioned in subsection 2.1 Brief introduction of the target office spaces to overview those target spaces, occupants, and on-site measurement processes. Please find the graphical floor plan in the attachment as a non-publish material.

*Section 2.3: please insert an extract of the questionnaire;

Respond to reviewer 3: The structure of questions with answering scales was addressed (line 264-281); and the extract of questionnaire was added as Figure 2, please find it in the revised manuscript.

*Table 4: please, explain how the authors calculate the values listed in the table;

Respond to reviewer 3: The preparation of data set was briefly mentioned in subsection 2.4 Classification of sample data and statistical analysis. Calculation process for Table 4 was added in line 343 - 346. For illuminance-based data, the horizontal illuminance at the desktop was derived from data logger (named as illuminance UV recorder, T&D TR-74) at the target times. Meanwhile, the vertical illuminance at the eye level was manually measured (by UVC light meter, type k/j thermometer, LX-200SD); it were then inputted as Ev terms into eq. 2 to calculate the DGPs values.

*Section 3.1, line 340: please, check the references.

Respond to reviewer 3: The number of references was changed to [28], which is refer to Chan et al. , 2015, to be referred as the threshold value of vertical illuminance at 2,670 lux.

If you need further information, please do not hesitate to contact me.
Your feedback will be highly appreciated. Thank you very much.

-------------------------------------------------------------------------------

References

  1. Hu, J.; Olbina, O. Illuminance-based slat angle selection model for automated control of split blinds. Environ. 2011, 46, 786-796.
  2. Mardaljevic, J. Examples of climate-based daylight modelling. In CIBSE National Conference 2006: Engineering the Future, London, UK, 2006.
  3. Reinhart, C.F.; Mardaljevic, J.; Rogers, Z. Dynamic daylight performance metrics for sustainable building design. 2006, 3(1), 1–25.
  4. Christoffersen, J.; Johnsen, K.; Petersen, E.; Valbjorn, O.; Hygge, S. Windows and daylight – A post-occupancy evaluation of Danish offices. In Proceedings CIBSE/ILE Joint Conference University of York, York, UK, 2000, 112–120.
  5. Wells, B.W.P. Subjective responses to the lighting installations in a modern office building and their design implications. Environ. 1965, 1, 57-68.
  6. Karlsen, L. R.; Heiselberg, P. K.; Bryn, I.; Johra, H.; Verification of simple illuminance based measures for indication of discomfort glare from windows. Environ. 2015, 92, 615–626.

Round 2

Reviewer 3 Report

Most of the concerns, suggestions and modifications requested by reviewers have been addressed.